# ScholarSum: Student-Teacher Abstractive Summarization via Knowledge Graph Reasoning and Reflective Refinement

## Abstract

Abstractive summarization of scientific papers is essential for efficient knowledge access. Although numerous approaches have been proposed, they often fail to capture the logical structure of the scientific paper, omit key factual information, and may produce hallucinated content. In this work, we propose ScholarSum, a Student–Teacher framework inspired by the human writing process, including drafting, reviewing, and revising. First, to capture paper structure, the student module constructs a knowledge graph based on the paper, divides it into semantic subgraphs, and performs graph-based reasoning to produce drafts aligned with the paper structure. Second, to improve coverage in long contexts, the student module retrieves key fact triplets from the global graph and integrates them into the draft, minimizing the loss of key factual information. Third, to strengthen factual fidelity, the teacher module conducts quality assessment via prompting and reference-guided reflection. Based on the assessment outcome, the module selects acceptance, minor revision, or regeneration. The collaborative design enables dynamic quality control, improving structural coherence and ensuring both factual completeness and accuracy. Experimental results on scientific summarization benchmarks demonstrate that ScholarSum consistently outperforms strong baselines, producing summaries that are structurally coherent, factually comprehensive, and well aligned with human-written reference summaries. Our code is available at `https://anonymous.4open.science/r/ScholarSum-Anonymous`.

## 1 Introduction

Scientific paper summarization plays a vital role in facilitating knowledge dissemination, reducing information overload, and supporting downstream research workflows. Unlike news or narrative texts, which often contain redundancy and follow relatively simple structures, scientific articles are typically organized according to the conventional IMRaD format consisting of Introduction, Methods, Results, and Discussion. A high-quality summary must therefore not only highlight the main findings but also preserve this logical structure, ensuring that claims are accurately linked to the supporting methods and results. In addition, scientific articles contain a high density of specialized information, which requires summaries to condense complex content without omitting details that are critical for correct interpretation. At the same time, factual precision is critical. Even minor inaccuracies or misrepresentations can distort a paper's contributions and mislead readers. Consequently, handling structural complexity, managing dense technical information, and ensuring factual consistency become the core points for high-quality scientific summarization (Gao et al., 2023; Cohan et al., 2018b; Xu & Lapata, 2020; Nan et al., 2021).

Over the past decades, the field of text summarization has evolved through several major technological shifts, which in turn have reshaped its core paradigms. Early research predominantly focused on extractive summarization, a paradigm where representative sentences are selected directly from the source text. This approach was powered by statistical and rule-based methods, including sentence scoring heuristics and graph-based algorithms like LexRank (Erkan & Radev, 2004). As the demand for more concise and human-like summaries grew, the focus shifted towards the more flexible abstractive paradigm,which aims to generate novel sentences that paraphrase and reorganize

the original content. This transition was largely enabled by the advent of deep learning, particularly with encoder-decoder models and pre-trained language models such as BERT (Devlin et al., 2018)and BART (Lewis et al., 2019), which significantly improved the fluency of generated text. The current era is dominated by the third wave of techniques, namely large language models (LLMs) like GPT-3 (Brown et al., 2020) and PaLM (Chowdhery et al., 2022). Pre-trained on massive corpora, these models demonstrate strong zero-shot and few-shot summarization capabilities, showing impressive adaptability across diverse domains without task-specific fine-tuning. Afterwards,

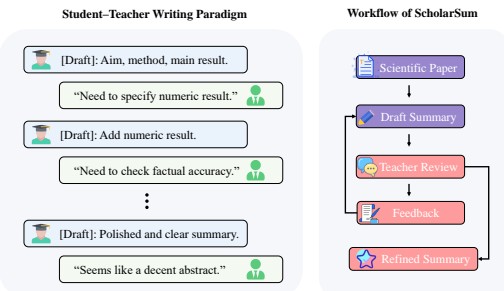

Figure 1: Illustration of ScholarSum. Left: human student–teacher writing process; Right: the workflow of ScholarSum inspired by the Left.

to address the persistent challenges of factual inconsistency and the static knowledge of standard LLMs, a fourth wave of techniques centered on Retrieval-Augmented Generation(RAG) has recently gained prominence. This paradigm enhances LLMs by first retrieving relevant documents or text passages from an external corpus before generating the summary, thereby grounding the model's output in verifiable information(Lewis et al., 2020). For scientific summarization, this approach is particularly effective as it can directly pull facts, figures, and methodological details from the source paper, significantly reducing the risk of factual hallucinations. Recently, this concept has been expanded to GraphRAG, which organizes the retrieved information using knowledge graphs (Edge et al., 2024). By constructing a comprehensive map of the document's concepts and their interconnections, this approach allows the model to integrate disparate information into a cohesive whole. This provides a broader perspective that is often absent in purely sequential processing.

Despite these advances, scientific paper summarization remains far from solved. First, the structural complexity of scientific articles challenges conventional summarization methods that process text sequentially. Sequential processing often fails to capture the logical structure and hierarchical relations across sections, for example, the linkage between methods and results (Cohan et al., 2018b). Second, even with RAG-based approaches, models often exhibit the "lost in the middle" phenomenon, where key evidence in long retrieved contexts is overlooked, leading to incomplete summaries (Liu et al., 2023). Third, factual fidelity remains a critical concern. Advanced LLMs are prone to generating content that is not supported by the source document, which is particularly detrimental in a scientific context (Gao et al., 2023). These limitations in preserving logical structure, handling long contexts, and ensuring factual fidelity emphasize the need for a more structured and controllable summarization framework.

To address these challenges, we propose ScholarSum, a novel Student-Teacher framework for scientific paper summarization. As shown in Figure 1, our approach is inspired by the collaborative human writing process, where a student drafts initial work, and a teacher provides guidance for iterative improvement. Analogous to how a student learns to write by receiving feedback from experienced teachers, our method splits the summarization task into two collaborative stages mirroring this natural pedagogical interaction. The student module first constructs a knowledge graph from the paper, clusters it into semantic subgraphs, and generates partial summaries using graph-based reasoning. To enhance coverage, key factual triples are retrieved from the graph and incorporated into the draft. The teacher module then evaluates the draft summary through a comprehensive critique, prompting, and reference-based reflection, offering targeted feedback to revise and improve the output iteratively. The collaborative design enables dynamic quality control, improving structural coherence and ensuring both factual completeness and accuracy. Our main contributions are summarized as follows:

- We propose ScholarSum, a novel Student-Teacher framework for the summarization of scientific papers, which is inspired by the process of human drafting and revision. The framework employs an iterative procedure, guided by quality ratings, to guide the cyclical process of drafting and refining.
- We introduce a novel abstraction strategy that leverages knowledge graph reasoning, which allows the student module to generate summaries that better reflect the document's structure and semantics with higher fidelity.

- We conduct extensive experiments on public scientific summarization benchmarks, demonstrating that our method significantly outperforms strong baselines in terms of structure, factuality, and human preference.

## 2 RELATED WORK

Our research intersects two key areas of study in abstractive summarization, both of which are particularly pertinent to the summarization of scientific and extensive documents. The first area focuses on the use of pre-trained models and iterative, reflective refinement techniques for summarization. The second area explores graph and knowledge graph-enhanced retrieval and generation methods that anchor summaries in structured evidence.

### 2.1 PRETRAINED MODELS AND ITERATIVE REFLECTIVE REFINEMENT

Abstractive summarization has moved from early sequence to sequence and pointer style architectures to large pretrained models. Pointer generator models and coverage mechanisms helped with copying and repetition (See et al., 2017). Pretraining objectives designed for summarization, such as the gap sentence objective and the unified text to text framework, established strong supervised baselines across many datasets (Zhang et al., 2020a; Raffel et al., 2020). More recent work shows how very large language models can be guided to produce better summaries using intermediate reasoning prompts, stepwise decomposition, or model distillation into smaller deployable systems (Wang et al., 2023; Xu et al., 2023).

Factuality is a central challenge in summarization, especially for scientific texts. Generated summaries must not present unsupported claims or misstate results. To reduce such errors, researchers have proposed iterative generation schemes that alternate between drafting, critique, verification, and revision. Empirical studies report that prompt chaining and multi step refinement often yield better scores and fewer factual errors than single pass prompting (Sun et al., 2024). Systems that use question answering style checks or targeted factuality signals can iteratively improve scientific summaries and reduce hallucination (Li et al., 2024b). These iterative methods motivate ScholarSum's student and teacher cycles for drafting and revision.

### 2.2 GRAPH AND RETRIEVAL AUGMENTED METHODS

Retrieval augmented generation is now a common way to ground text generation in external evidence. When the evidence has relational structure, such as citation links or discourse relations, graph aware retrieval and graph aware generation can better capture document level relations that matter for coherent and faithful summaries. Recent surveys and system papers describe pipelines that combine graph based indexing, subgraph retrieval, and graph informed generation and they note specific challenges for textual graphs and citation networks (Peng et al., 2024).

In the fields of scientific and extensive document summarization, graph-based methods are employed to dissect documents into cohesive subtopics. These methods facilitate the retrieval of evidence as subgraphs, rather than as isolated passages, and aid in guiding the generation process, ensuring adherence to entity and relation structure. Studies on plan-guided and graph-constrained planning demonstrate that self-correcting planning and graph-constrained decoding assist in maintaining reasoning in alignment with the underlying graph structure (Chen et al., 2024; Li et al., 2024a). Frameworks that combine graph retrieval with generation have been shown to reduce instances of hallucination and enhance support for multi-hop document reasoning, an essential aspect of scientific summaries (Hu et al., 2025; Peng et al., 2024).

Prior work suggests three complementary components for high quality scientific summaries. First, strong pretrained generative models provide fluent abstraction. Second, iterative critique and revision improve factuality and coherence. Third, graph grounded retrieval supplies verifiable evidence and discourse structure. ScholarSum combines these components by building document derived graphs, performing subgraph aware drafting at the student level, and applying teacher level reflective feedback that is grounded in retrieved evidence.

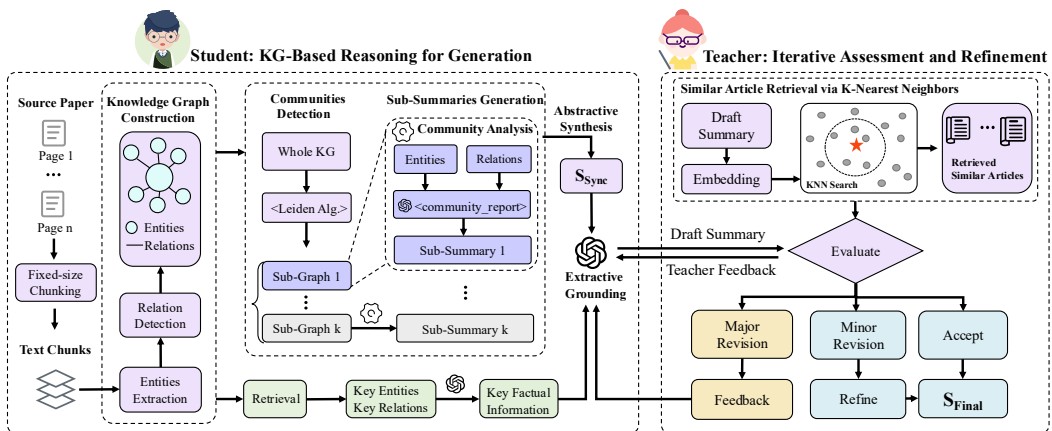

Figure 2: Our student-teacher framework. The student (left) generates a draft, which the teacher (right) iteratively refines with feedback from retrieved exemplars.

## 3 METHODOLOGY

In this section, we first formalize the problem definition of scientific summarization. We then provide an overview of the ScholarSum framework. Next, we describe the student module, including knowledge graph construction, semantic subgraph partitioning, and graph based reasoning, followed by the teacher module for assessment and feedback driven revision. We also explain the iterative refinement schedule and the stopping criterion used in our framework.

### 3.1 PROBLEM DEFINITION

Given a scientific paper $D$, the goal of abstractive summarization is to produce a coherent summary $S$ that faithfully captures the principal contributions and preserves the document's logical flow.

### 3.2 OVERVIEW

Figure 2 provides the framework of ScholarSum, which consists of two synergistic modules operating in an iterative loop. The student module constructs a knowledge graph from the source paper, decomposes it into semantically coherent subgraphs, and produces a draft summary via graph-based reasoning, further supplementing coverage by retrieving salient factual triples. In response, the teacher module critically reviews the draft using both prompting and retrieval-based evaluation and then issues feedback in one of three forms: acceptance, minor revision, or major revision. This cycle of proposal and critique continues until a termination condition is met, steering the student toward increasingly refined outputs. We next describe the two modules in detail.

### 3.3 STUDENT: KG-BASED REASONING FOR GENERATION

The Student module serves as the generative engine of the framework, producing a coherent summary that is firmly grounded in structured knowledge extracted from the source document. Its workflow is deliberately crafted to balance two often competing objectives in summarization: *abstractive fluency* and *factual fidelity*, ensuring that the final content remains both fluent and reliably accurate.

**Structured Knowledge Representation.** Our methodology begins with the fundamental step of transforming the unstructured text of a given source paper $D$, into a structured and machine-readable format. To achieve this, we construct a Knowledge Graph (KG), formally denoted as $G = (V, E)$. This graph serves as a semantic blueprint of the paper's core content.

The vertices $V$ in the graph represent the essential scientific entities discussed in the paper. We categorize these entities into predefined, high-level concepts that are central to scientific discourse, including *Tasks*, *Methods*, *Metrics*, and *Datasets*. By identifying and isolating these key components, we lay the groundwork for a deeper, more structured understanding of the paper's contributions.

The edges, $E$, of the graph represent the rich semantic relationships that exist between these entities, effectively encoding the interdependencies described in the text. For instance, a relation can link a *Method* to a *Task* it is designed to solve, connect a *Method* to the *Metric* it aims to improve, or associate a *Task* with the *Dataset* used for its evaluation. These relational links are crucial as they capture the logical flow and experimental setup of the research. This process converts the linear narrative of the paper into a highly organized semantic map. This structured representation is not only machine-interpretable but also provides a robust and explicit foundation for the subsequent reasoning and content generation stages of our framework.

**Thematic Segmentation via Community Detection.** Scientific articles often weave multiple thematic threads. To algorithmically surface these threads, we apply the Leiden algorithm (Traag et al., 2019) to $G$, yielding $k$ semantic subgraphs, or communities:

$$\{G_1, G_2, \ldots, G_k\} = \text{Leiden}(G), \tag{1}$$

where each $G_i$ representing a well-defined sub-topic (e.g., background, methodology) within the study. The Leiden method identifies groups that are dense internally and sparse across different groups. It is efficient and capable of scaling to large graphs, providing stable and high-quality partitions. The resulting thematic segmentation serves as a coherent foundation for subsequent analytical procedures. Additionally, we filter out very small communities and merge highly similar ones to enhance robustness.

**Two-Stage Summary Generation.** At the core of the student module lies a two-stage generation process, which explicitly integrates *abstractive fluency* with *factual grounding*. The first stage emphasizes coherent narrative construction, while the second stage ensures that this narrative is supported by verifiable, fine-grained details.

*Abstractive synthesis:* For each thematic subgraph $G_i$ discerned in the preliminary clustering phase, a substantial language model generates a succinct sub-summary $s_i$, encapsulating the critical contribution of that subgraph. Subsequently, these sub-summaries are amalgamated in accordance with their thematic sequence to construct an initial draft.

$$S_{\text{draft}} = s_1 \oplus \cdots \oplus s_k,$$

where $s_i$ denotes the concise sub-summary generated for the $i$-th thematic subgraph $G_i$, and $\oplus$ represents the concatenation operator that assembles the sub-summaries in their thematic order. The resulting $S_{\text{draft}}$ serves as an initial, coverage-oriented summary that preserves the global topical structure of the source document, acting as a scaffold for the subsequent grounding phase.

*Extractive grounding:* To ensure factual accuracy, we commence by extracting key factual triplets from the principal knowledge graph $G$. This extraction process is steered by domain-specific logical keywords, which aid us in identifying and extracting triplets that depict crucial entities and their relationships, such as specific dataset identifier and key numerical results. The aggregation of these triplets forms a focused context subgraph, denoted as $G_{\text{context}}$. Subsequently, this subgraph, the draft summary $S_{\text{draft}}^{(i)}$, and the teacher feedback $F_T^{(i-1)}$ are provided as inputs to a large language model. The model employs chain-of-thought reasoning to refine the draft and generate an updated, factually accurate student summary:

$$S_{\text{student}}^{(i)} = \mathcal{F}_{\text{CoT}}(S_{\text{draft}}^{(i)}, G_{\text{context}}, F_T^{(i-1)}), \tag{2}$$

where $F_T^{(0)}$ is null for the initial pass. This stage integrates factual anchors from the context subgraph into the abstractive draft while also correcting inaccuracies identified in earlier iterations. In this way, logical keywords act as a bridge between structured evidence and iterative refinement, ensuring that the student module consistently produces summaries that are both coherent and well grounded.

By explicitly separating narrative synthesis from fact insertion, this two-stage design enables the student module to generate summaries that read naturally while maintaining rigorous adherence to the source material, achieving a balance that single-stage approaches often fail to.

### 3.4 TEACHER: ITERATIVE ASSESSMENT AND REFINEMENT

The teacher module, acting as a discriminator, evaluates the student's output and guides its refinement. This paradigm is analogous to reinforcement learning-based summarization, where decoupling generation from assessment fosters more stable and goal-aligned outputs (Paulus et al., 2017).

This separation ultimately enhances summarization quality, reliability, and reproducibility across diverse scientific domains.

**Quality Evaluation Module.** The teacher first measures the quality of $S_{\text{student}}^{(i)}$ using both quantitative and qualitative lenses. A K-Nearest Neighbors (KNN) search retrieves $k$ similar papers $\{D_j'\}_{j=1}^k$ and their abstracts $\{A_j'\}_{j=1}^k$ from a reference corpus, providing a domain-relevant benchmark for comparison. The evaluation function $\mathcal{G}_{\text{evaluate}}$ then computes:

$$\sigma^{(i)}, F_T^{(i)} = \mathcal{G}_{\text{evaluate}}(S_{\text{student}}^{(i)}, \{A_j'\}_{j=1}^k), \tag{3}$$

where $\sigma^{(i)}$ denotes a scalar quality score, while $F_T^{(i)}$ symbolizes a structured set of feedback items. These items are derived from a comparison with reference abstracts, thereby providing a foundation for subsequent refinement steps.

**Revision Action Notifier.** The Notifier, guided by $\sigma^{(i)}$, determines the subsequent step in the revision process, thereby more effectively minimizing unnecessary edits that may otherwise stem from vague or underspecified revision prompts. The decision mechanism is governed by two thresholds, $\theta_{\text{major}}$ and $\theta_{\text{minor}}$, where $\theta_{\text{minor}} \geq \theta_{\text{major}}$.

*Accept and Minor Revisions:* If the value of $\sigma^{(i)}$ is greater than or equal to $\theta_{\text{minor}}$, then the summary is accepted directly. And if $\sigma^{(i)}$ is less than $\theta_{\text{minor}}$ but greater than $\theta_{\text{major}}$, it will be accepted after minor revision by the teacher:

$$S_{\text{final}} = \mathcal{F}_{\text{minor\_rev}}(S_{\text{student}}^{(i)}, F_T^{(i)}), \tag{4}$$

where $\mathcal{F}_{\text{minor\_rev}}$ is a function that implements minor revisions based on the feedback $F_T^{(i)}$, and $S_{\text{final}}$ symbolizes the final, publication-ready summary that fulfills the quality threshold.

*Request for Major Revision:* If the value of $\sigma^{(i)}$ is less than or equal to $\theta_{\text{major}}$, the teacher will supply the student with $F_T^{(i)}$ for the subsequent iteration. This provision offers structured guidance on the important content and necessary organizational modifications.

$$\text{Utilize } F_T^{(i)} \text{ in Eq.(2) for iteration } i + 1.$$

This iterative process continues until the output satisfies the quality criteria or a predetermined iteration limit is reached, thereby balancing improvement with efficiency.

Through this structured interaction between student and teacher, ScholarSum consistently improves draft quality while ensuring efficiency and consistency across iterations. *The pseudocode for ScholarSum is provided in Appendix A.*

## 4 EXPERIMENTS

We conduct extensive experiments to rigorously evaluate the efficacy and robustness of ScholarSum. Our results across diverse benchmarks highlight its strong generalization ability and consistent improvements over competitive baselines.

### 4.1 EXPERIMENTAL SETUP

**Datasets and Metrics.** We evaluate ScholarSum on two widely used scientific summarization benchmarks: *ArXiv* and *PubMed* (Cohan et al., 2018a). For evaluation, we report ROUGE (R-1, R-2, R-L) (Lin, 2004) for lexical overlap, METEOR (Banerjee & Lavie, 2005) for semantic similarity, and BERTScore (Zhang et al., 2020b) for contextual semantic alignment. *Detailed dataset statistics and implementation settings are provided in Appendix B.*

**Baselines.** We compare against two groups of competitive baselines: (1) Traditional encoder–decoder summarization models: T5 (Raffel et al., 2020), LED (Beltagy et al., 2020), and PEGASUS (Zhang et al., 2020a); (2) LLM-based prompting methods: SumCot (Wang et al., 2023) and QA-prompting (Sinha, 2025), evaluated with DeepSeek and Qwen base models.

Table 1: Main experimental results on the ArXiv and Pubmed datasets. Best results in each column are highlighted in **bold**, and second-best are underlined.

| Dataset | Base LLM | Models | R-1 | R-2 | R-L | METEOR | BERTScore |
|---|---|---|---|---|---|---|---|
| **ArXiv** | None | T5 | 0.2638 | 0.0670 | 0.2323 | 0.1587 | 0.8273 |
| | | LED | 0.2267 | 0.0605 | 0.2000 | 0.1972 | 0.7739 |
| | | PEGASUS | 0.2550 | 0.0626 | 0.2034 | 0.1597 | 0.8131 |
| | DeepSeek | SumCot | 0.2027 | 0.0409 | 0.1837 | 0.2230 | 0.8128 |
| | | QA-prompting | 0.2635 | 0.0694 | 0.2312 | 0.2362 | 0.8294 |
| | | **Ours** | 0.2692 | **0.0708** | 0.2362 | 0.2300 | **0.8360** |
| | Qwen | SumCot | 0.1940 | 0.0390 | 0.1730 | 0.2456 | 0.8133 |
| | | QA-prompting | 0.2339 | 0.0652 | 0.2097 | 0.2539 | 0.8154 |
| | | **Ours** | **0.2764** | 0.0646 | **0.2412** | **0.2541** | 0.8338 |
| **PubMed** | None | T5 | 0.2560 | 0.0809 | 0.2345 | 0.1427 | 0.8253 |
| | | LED | 0.2447 | 0.0739 | 0.2211 | 0.2100 | 0.7808 |
| | | PEGASUS | 0.2512 | 0.0687 | 0.2172 | 0.1364 | 0.8167 |
| | DeepSeek | SumCot | 0.1934 | 0.0299 | 0.1801 | 0.1818 | 0.8141 |
| | | QA-prompting | 0.2585 | 0.0663 | 0.2325 | 0.2187 | 0.8394 |
| | | **Ours** | **0.3102** | **0.0928** | **0.2834** | **0.2567** | **0.8531** |
| | Qwen | SumCot | 0.2060 | 0.0412 | 0.1851 | 0.2312 | 0.8191 |
| | | QA-prompting | 0.2634 | 0.0748 | 0.2410 | 0.2496 | 0.8316 |
| | | **Ours** | 0.2929 | 0.0735 | 0.2645 | 0.2505 | 0.8435 |

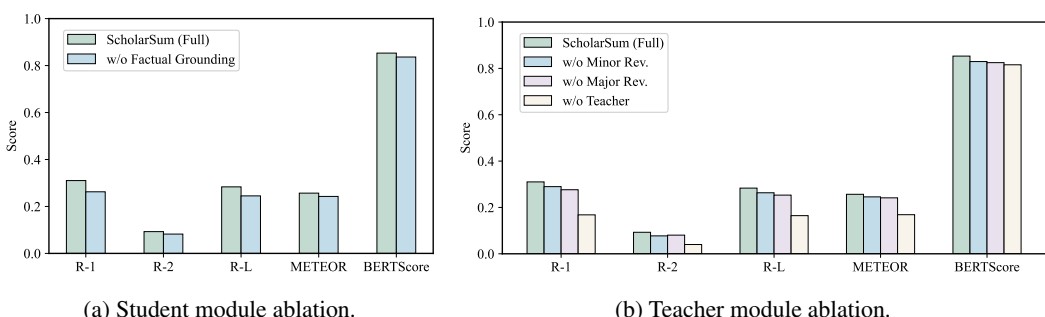

(a) Student module ablation.     (b) Teacher module ablation.

Figure 3: Analysis of component contributions via ablation studies.

## 4.2 RESULTS ANALYSIS

**Main Results Analysis.** Table 1 presents the principal quantitative outcomes. ScholarSum maintains consistent strong performance across both datasets and all evaluation metrics, surpassing traditional encoder and decoder summarization models, as well as prompt-based large language model approaches. On PubMed, where biomedical abstracts are dense and laden with terminology, ScholarSum delivers particularly significant enhancements. With DeepSeek, it improves ROUGE-1 by +19.9% and METEOR by +17.4% over the most robust baseline. These gains suggest substantially enhanced factual coverage and semantic fidelity. Similar improvements are observed under the Qwen framework, demonstrating the framework's robustness across various LLM architectures.

The observed enhancements are attributed to two primary design decisions. Firstly, reasoning based on a knowledge graph provides a structured discourse-level understanding of the source text. Secondly, the reflective refinement within the teacher module iteratively enhances coherence and factual accuracy by providing context-aware, targeted feedback. Although the standard T5 model achieves competitive ROUGE scores, its lower METEOR and BERTScore indicate limitations in paraphrase handling and deeper semantic alignment. ScholarSum is specifically designed to address these limitations in a principled and systematic manner. Moreover, we find that the method improves summary consistency and diminishes unsupported statements. These practical advantages make the framework well-suited for real-world scientific summarization tasks.

Table 2: Analysis of hyperparameter sensitivity.

| Temperature | R-1 | R-2 | R-L | METEOR | BS |
|---|---|---|---|---|---|
| 1.0 | 0.2880 | 0.0806 | 0.2650 | 0.2472 | 0.8484 |
| **0.8** | **0.3102** | **0.0928** | **0.2834** | **0.2567** | **0.8531** |
| 1.3 | 0.2869 | 0.0801 | 0.2614 | 0.2421 | 0.8487 |
| 0.2 | 0.2789 | 0.0713 | 0.2550 | 0.2262 | 0.8452 |

(a) Effect of temperature on generation quality.

| Keyword Ver. | R-1 | R-2 | R-L | METEOR | BS |
|---|---|---|---|---|---|
| task | 0.2835 | 0.0804 | 0.2545 | 0.2446 | 0.8469 |
| **query1** | 0.3093 | 0.1048 | **0.2929** | **0.2575** | **0.8528** |
| query2 | 0.3072 | **0.1082** | 0.2840 | 0.2515 | 0.8501 |
| query3 | 0.3014 | 0.0978 | 0.2883 | 0.2473 | 0.8515 |
| LLM-adaptive | **0.3098** | 0.0979 | 0.2909 | 0.2529 | 0.8454 |

(b) Comparison of logical keyword strategies.

**Ablation Studies Analysis.** To assess the contributions of components within both the student and teacher modules, we perform systematic ablation experiments. The results emphasize the importance of each module and its subcomponents for producing high quality summarization results.

**Student Ablation:** Figure 3a shows that removing the factual extractive grounding module (*w/o Factual Grounding*) leads to consistent drops across all metrics, confirming that anchoring the generation process to extracted evidence plays a pivotal role in ensuring factual accuracy.

**Teacher Ablation:** Figure 3b illustrates that the removal of the teacher module results in the most significant performance degradation, underscoring the pivotal role of reflective revision. Among the subcomponents, the Major Revision stage exerts a greater influence than the Minor Revision stage, suggesting that high-level structural critique is particularly valuable.

**Hyperparameter Sensitivity Analysis.** We analyze how two key hyperparameters affect generation quality: decoding temperature and logical keyword selection strategy.

As indicated in Table 2a, a temperature setting of 0.8 provides the optimal trade-off between diversity and factual consistency. Compared to a temperature setting of 1.0, the configuration of 0.8 yields higher scores across all the metrics. Lower temperatures, such as 0.2, make the model overly conservative and decrease variation in the output. On the other hand, higher temperatures, such as 1.3, increase randomness and result in declines in metric scores and occasional incoherent sentences. For these reasons, we adopt a temperature setting of 0.8 as the default in our experiments, as it enhances overall quality while maintaining a low rate of factual errors.

Table 2b compares different keyword selection strategies. Structured prompts (Query1–3) achieve better results than unguided generation, showing that logical scaffolding improves summary coherence and factual accuracy. In our framework, logical keywords guide the Extractive Grounding stage by identifying a context subgraph with key factual anchors such as dataset names and numerical results. This subgraph allows the model to supply missing details

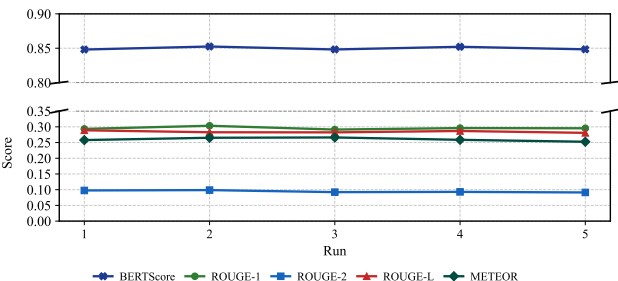

Figure 4: Analysis of model performance stability.

and correct errors, explaining the effectiveness of structured prompts. Although the LLM-adaptive method has yet to surpass manual strategies, its ability to incorporate contextual signals suggests promise for future improvement.

**Stability Analysis.** To evaluate the performance stability of our framework, we conducted five independent runs using different random seeds. The results, illustrated in Figure 4, reveal a high degree of consistency across all evaluation metrics. Specifically, the standard deviation for ROUGE scores is remarkably low (e.g., $\approx 0.004$ for ROUGE-L), with similarly negligible variance observed for METEOR and BERTScore. This minimal fluctuation demonstrates that the model's performance is not an artifact of stochasticity but rather a deterministic outcome of its structured methodology, underscoring its robustness for practical applications.

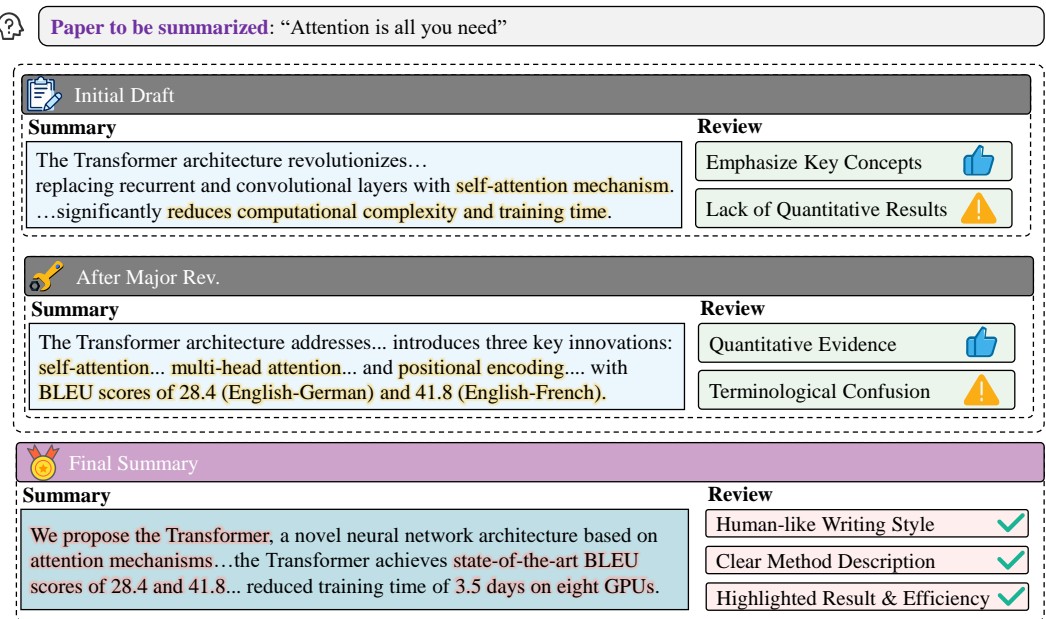

Figure 5: An illustration of ScholarSum's iterative refinement process on the abstract of "Attention Is All You Need". The figure showcases the summary's evolution at each stage.

## 4.3 CASE STUDY ANALYSIS

To qualitatively illustrate our iterative refinement process, we present a case study on the influential paper "Attention Is All You Need" (Vaswani et al., 2017). Figure 5 shows the summary's evolution across multiple revision stages, highlighting the improvements achieved at each step.

The process commences with an Initial Draft, which, while technically accurate and incorporating essential terminology, lacks a cohesive argumentative structure. Under the guidance of our teacher module, the Major Revision restructures the narrative into a clear problem-solution format and integrates key statistical context, thereby enhancing both readability and logical flow. The Final Summary then undergoes further refinement for conciseness and rhetorical impact, such as adopting active phrasing like "We propose the Transformer...". This brings it in line with the stylistic conventions of the original Ground Truth abstract. This progression illustrates how ScholarSum iteratively enhances both factual accuracy and the overall rhetorical quality of its generated summaries.

## 5 CONCLUSION

We introduce ScholarSum, a student-teacher system designed to summarize scientific papers. By integrating knowledge graphs and a review and correction cycle, the system generates high-quality summaries. ScholarSum operates in two phases. Initially, a student module reads the paper, constructs basic knowledge maps, and drafts an initial summary, ensuring the main ideas and key terms are captured. Subsequently, a teacher module, acting as an expert, examines the summary. It provides specific improvement suggestions using intelligent prompts and by identifying correct information to rectify errors. This review and improvement cycle enhances the summary's clarity, accuracy, and completeness. Our experiments demonstrate that ScholarSum performs exceptionally well, surpassing other leading methods. The summaries it produces are well-organized, factually correct, and closely resemble those written by humans. This study underscores the value of employing structured thinking and iterative feedback for summary creation. For future work, we aim to expand ScholarSum to more scientific fields and incorporate information from figures and tables.

## 6 ETHICS STATEMENT

We confirm that this work aligns with accepted ethical standards in machine learning research. All data and methodologies used are publicly available or properly cited.

## 7 REPRODUCIBILITY STATEMENT

To support reproducibility, we have provided full details of our experimental setup, including hyper-parameters and dataset descriptions, in the experimental section. Code is available.

## 8 THE USE OF LARGE LANGUAGE MODELS (LLMS)

We utilize LLMs to assist and enhance our writing. They help us improve the quality and effectiveness of our textual expression.

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

# A ScholarSum: Algorithmic Details

This section provides a detailed algorithmic description of the ScholarSum framework. Our approach is centered around an iterative student-teacher architecture, designed to progressively refine a summary until it meets a predefined quality threshold. The core components are the main inference loop, the student module for generation, and the teacher module for assessment.

## A.1 Main Inference Loop

The main inference process of the ScholarSum framework is detailed in Algorithm 1. It begins by constructing a knowledge graph from the source document, then enters an iterative loop where the Student module generates a summary, and the Teacher module evaluates it. The process terminates either when the summary quality is sufficient for a minor revision or when the maximum number of iterations is reached.

---

**Algorithm 1** ScholarSum: Main Inference Loop

**Require:** Source document $D$, knowledge graph builder, quality thresholds $\theta_{\text{minor}}$, $\theta_{\text{major}}$, and max iterations $I_{\text{max}}$.

1: $G \leftarrow \text{CONSTRUCTKG}(D)$
2: $F_T^{(0)} \leftarrow \text{NULL}$
3: **for** $i = 1$ **to** $I_{\text{max}}$ **do**
4: $\quad S_{\text{student}}^{(i)} \leftarrow \text{STUDENTMODULE}(D, G, F_T^{(i-1)})$
5: $\quad \sigma^{(i)}, F_T^{(i)}, S_{\text{final}} \leftarrow \text{TEACHERMODULE}(S_{\text{student}}^{(i)})$
6: $\quad$ **if** $S_{\text{final}} \neq \text{NULL}$ **then**
7: $\quad\quad$ **return** $S_{\text{final}}$
8: $\quad$ **end if**
9: **end for**
10: **return** $S_{\text{student}}^{(I_{\text{max}})}$

---

## A.2 Student Module

The Student module, described in Algorithm 2, is responsible for generating the summary draft. It first partitions the global knowledge graph into coherent sub-graphs or communities. These communities are then synthesized into an initial draft summary. Finally, it refines this draft using a Chain-of-Thought (CoT) reasoning process, which incorporates contextual information retrieved from the KG and any feedback from the previous iteration's Teacher evaluation.

---

**Algorithm 2** StudentModule

---

**Require:** Document $D$, Knowledge Graph $G$, and Teacher Feedback $F_{\text{feedback}}$.

1: $\{G_1, \ldots, G_k\} \leftarrow \text{CLUSTERGRAPHS}(G)$
2: $S_{\text{draft}} \leftarrow \text{ABSTRACTIVESYNTHESIS}(G_1, \ldots, G_k)$
3: $G_{\text{context}} \leftarrow \text{RETRIEVECONTEXT}(G, \text{KEYWORDS})$
4: $S_{\text{student}} \leftarrow \mathcal{F}_{\text{CoT}}(S_{\text{draft}}, G_{\text{context}}, F_{\text{feedback}})$
5: **return** $S_{\text{student}}$

---

### A.3 TEACHER MODULE

The Teacher module (Algorithm 3) acts as the quality gate. It assesses the student-generated summary $S_{\text{student}}^{(i)}$ to produce a quality score $\sigma^{(i)}$ and structured feedback $F_T^{(i)}$. Based on this score, it makes a three-way decision: (1) accept the summary with minor revisions if it exceeds $\theta_{\text{minor}}$, (2) trigger another iteration with detailed feedback for a major revision if the score is below $\theta_{\text{major}}$, or (3) perform a light refinement and re-evaluate if the quality is moderate.

---

**Algorithm 3** TeacherModule

---

**Require:** Student summary $S_{\text{student}}^{(i)}$, and thresholds $\theta_{\text{major}}, \theta_{\text{minor}}$.

1: $(\sigma^{(i)}, F_T^{(i)}) \leftarrow \mathcal{G}_{\text{assess}}(S_{\text{student}}^{(i)})$
2: **if** $\sigma^{(i)} < \theta_{\text{minor}}$ and $\sigma^{(i)} > \theta_{\text{major}}$ **then**
3: $\quad S_{\text{final}} \leftarrow \mathcal{F}_{\text{minor\_rev}}(S_{\text{student}}^{(i)}, F_T^{(i)})$
4: $\quad$ **return** $(\sigma^{(i)}, F_T^{(i)}, S_{\text{final}})$
5: **else if** $\sigma^{(i)} \leq \theta_{\text{major}}$ **then**
6: $\quad$ **return** $(\sigma^{(i)}, F_T^{(i)}, \text{NULL})$
7: **else**
8: $\quad S_{\text{final}} \leftarrow S_{\text{student}}^{(i)}$
9: $\quad$ **return** $(\sigma^{(i)}, F_T^{(i)}, S_{\text{final}})$
10: **end if**

---

## B REPRODUCIBILITY

To ensure the reproducibility of our results, this section details the datasets, hyperparameters, models, and hardware used in our experiments.

### B.1 DATASETS AND PREPROCESSING

We evaluate our framework on two widely-used and challenging long-document summarization benchmarks: ArXiv and PubMed. These datasets are composed of full-length scientific articles, making them ideal for assessing a model's ability to handle lengthy and complex texts. The ArXiv dataset consists of papers

Table 3: Descriptive statistics for the ArXiv and PubMed long-document summarization datasets.

| Dataset | Split | # Docs | Avg. Doc. Len. | Avg. Sum. Len. |
|---|---|---|---|---|
| ArXiv | 203K/6.4K/6.4K | 215K | $\approx 6{,}040$ | $\approx 231$ |
| PubMed | 119K/6.6K/6.7K | 133K | $\approx 3{,}025$ | $\approx 203$ |

from physics, computer science, and mathematics, while PubMed focuses on biomedical literature. A key characteristic of these benchmarks is that the ground-truth summaries are typically the author-written abstracts, which serve as high-quality, human-generated references. The primary challenges they present include the sheer document length and the necessity for models to understand highly technical language and capture long-range dependencies between different sections of a paper, such as connecting the introduction to the conclusions. Table 3 provides a detailed statistical overview of these datasets. The data splits for train, validation, and test sets are noted respectively.

Table 4: Hyperparameter settings for the ScholarSum framework.

| Hyperparameter | Value |
| --- | --- |
| *Framework Control* | |
| Max Iterations ($I_{\max}$) | 5 |
| Major Revision Threshold ($\theta_{\mathrm{major}}$) | 0.60 |
| Minor Revision Threshold ($\theta_{\mathrm{minor}}$) | 0.85 |
| *Model & Generation* | |
| k for kNN (Teacher Module) | 5 |
| Generation Temperature | 0.8 |
| *Logical Keyword Queries* | |
| Query 1 | study design, methodology, key findings, implications, limitations |
| Query 2 | background, objectives, methods, results, conclusions, future work |
| Query 3 | research question, experimental approach, main outcomes, relevance |

## B.2 HYPERPARAMETER CONFIGURATION

The key hyperparameters for ScholarSum were determined through systematic grid search and are outlined in Table 4. These settings were kept consistent across all experiments to ensure a fair comparison.

## B.3 MODELS EVALUATED

To benchmark ScholarSum, we compare it against a suite of powerful large language models and established summarization baselines. The primary models are:

- **DeepSeek-V3**: A 671B parameter Mixture-of-Experts (MoE) model.
- **Qwen2.5-Turbo**: An efficient and powerful model optimized for speed.

We also include the following traditional summarization baselines:

- GOOGLE-T5/T5-LARGE
- ALLENAI/LED-LARGE-16384
- GOOGLE/PEGASUS-LARGE

## B.4 COMPUTING INFRASTRUCTURE

All experiments were conducted on a high-performance computing cluster equipped with NVIDIA Tesla V100 Tensor Core GPUs.

## C CORE PROMPTS FOR SCHOLARSUM

The performance of LLM-based frameworks heavily depends on the quality of the prompts. For full transparency, we provide in this section the exact prompts that guide the behavior of ScholarSum.

### C.1 COMMUNITY SUMMARY INTEGRATION PROMPT

The following prompt is used by the Student module to synthesize multiple community-level summaries, which are generated from different clusters of the knowledge graph, into a coherent draft summary. The prompt emphasizes integration, coherence, and adherence to factual information present in the provided reports.

***Prompt: Summary Integration***

**Role:** You are a helpful assistant synthesizing multiple sub-summaries into a coherent comprehensive summary.

**Goal:** Generate a response of the target length and format that integrates multiple sub-summaries from analysts who focused on different parts of the dataset into a unified summary.

Note that the analysts' reports provided below are ranked in the **descending order of importance**.

If you don't know the answer or if the provided reports do not contain sufficient information to provide an answer, just say so. Do not make anything up.

The final response should remove all irrelevant information from the analysts' reports and merge the cleaned information into a comprehensive summary that provides explanations of all the key points and implications appropriate for the response length and format.

Add sections and commentary to the response as appropriate for the length and format. Style the response in markdown.

The response shall preserve the original meaning and use of modal verbs such as "shall", "may" or "will".

Do not include information where the supporting evidence for it is not provided.

**Target response length and format:** {response_type}

**Analyst Reports:** {report_data}

## C.2    EXTRACTIVE GROUNDING PROMPT

This prompt guides the Chain-of-Thought reasoning process within the Student module. It instructs the model to ground the abstractive draft summary with concrete details retrieved from the knowledge graph, ensuring the final output is both comprehensive (globally) and accurate (locally).

*Prompt: Extractive Grounding*

**Role:** You are an expert research assistant synthesizing information from multiple sources to answer a query comprehensively using a step-by-step reasoning process.

**Query:** {query}

**Global Insights (Summary of Key Points from Community Reports):** {global_points_context}

**Detailed Local Context (Entities, Relationships, Sources):** {local_context}

**Task:**

1. **Analyze the Query:** Briefly restate the main goal of the query: {query}
2. **Synthesize Globally:** Based on the "Global Insights", what are the main high-level takeaways relevant to the query?
3. **Synthesize Locally:** Based on the "Detailed Local Context", what specific entities, relationships, or source details provide evidence or examples related to the query and the global takeaways?
4. **Chain of Thought Reasoning:** Explain step-by-step how you will combine the global perspective and local details to construct the final answer. Bridge the high-level findings with specific evidence.
   - Start with the global context.
   - Use local details to elaborate, support, or nuance the global points.
   - Ensure all aspects of the original query are addressed.
5. **Final Comprehensive Answer:** Based on your reasoning, provide a final, coherent response of type {response_type} that directly answers the query, integrating both global perspectives and specific local details.

**Reasoning Steps (Chain of Thought):** ⟨Your step-by-step reasoning process goes here⟩

**Final Answer:** ⟨Your final synthesized answer of type {response_type} goes here⟩

## C.3    TEACHER EVALUATION PROMPT

The Teacher module operates based on the following prompt, which defines its persona as a hyper-critical expert. This prompt enforces a rigorous, multi-faceted evaluation of the student's summary against a strict set of criteria, from structural compliance to scientific accuracy, and requires structured, actionable feedback.

*Prompt: Teacher Evaluation*

**Role:** Hyper-Critical Scientific Abstract Evaluation Expert with Extreme Academic Rigor

**Objective:** Conduct a comprehensive, systematic, and uncompromisingly precise evaluation of the scientific abstract.

**Absolute Evaluation Criteria:**

**1. Structural Compliance (Non-Negotiable)**

- MANDATORY: Abstract MUST be a SINGLE, COHESIVE PARAGRAPH
- Total word count is limited to 200-250 words, no less or more.
- Immediate critical assessment of paragraph structure and coherence

**2. Background and Significance**

- Demand SURGICAL-LEVEL clarity of scientific context
- Instantaneous and precise identification of knowledge gap
- Zero tolerance for vague or generalized contextual statements

**3. Research Objectives**

- Precisely defined, Unambiguously measurable, Directly traceable to background context

**4. Methodology Scrutiny**

- Forensic-level precision, Explicit justification of each methodological approach, Unequivocal alignment with research objectives, Demand comprehensive yet concise methodological explanation

**5. Results and Implications**

- Statistical significance, Direct correlation to initial objectives, Quantitative precision, Implications must extend beyond immediate findings

**6. Technical Considerations**

- Crisp and active, Devoid of unnecessary jargon, Scientifically precise, Logical and coherent structure mandatory

**Comparative Analysis:**

- Rigorously compare the generated abstract with GROUND TRUTH reference papers
- Assess: Content alignment, Scientific accuracy, Presentation style coherence
- Identify ANY deviations or potential inaccuracies

**Evaluation Output Format:**

If the abstract meets standards:

```
Precision Score: [Numerical score/100]
```

If the abstract requires revisions:

```
Precision Score: [Numerical score/100]
Improvement Suggestions:
- [Actionable suggestion 1]
- [Actionable suggestion 2]
...
```

**Submission Materials:**

- Generated Abstract: {summary}
- Original Article: {article}
- Reference Papers (GROUND TRUTH): {ref_papers}

**Mandate:** Provide a comprehensive, nuanced, and ruthlessly precise scholarly evaluation.

