# OpenReview forum: "ScholarSum: Student-Teacher Abstractive Summarization via Knowledge Graph Reasoning and Reflective Refinement"
_ICLR.cc/2026/Conference — ICLR 2026 Conference Withdrawn Submission_

### Official Review · Reviewer_VAgV · 2025-10-20

**Soundness:** 2
**Presentation:** 2
**Contribution:** 2
**Rating:** 4
**Confidence:** 4

**Summary:**

The paper introduces ScholarSum, a two-module, iterative framework for scientific paper summarization. The Student module builds a paper-specific knowledge graph (entities: Tasks/Methods/Metrics/Datasets; edges encode relations), partitions it with Leiden, drafts sub-summaries per community, and performs “extractive grounding” by inserting factual KG triplets. The Teacher module retrieves similar papers via k-NN, scores the draft, and issues accept / minor / major revision feedback until quality thresholds are met or iteration limits are reached. On arXiv and PubMed, ScholarSum outperforms T5/LED/PEGASUS and LLM-prompting baselines (SumCoT, QA-prompting with DeepSeek and Qwen) across ROUGE, METEOR, and BERTScore. Ablation results indicate that both KG-level grounding and the Teacher loop materially contribute. Prompts, hyperparameters, and pseudocode are included for reproducibility.

**Strengths:**

- A clean and well-structured integration of KG-guided planning (Leiden communities → sub-summaries) with a retrieval-backed Teacher for targeted iteration; the accept/minor/major revision policy is simple and implementable.
- Strong performance on arXiv/PubMed across ROUGE, METEOR, and BERTScore; ablations isolate the contribution of KG grounding and the Teacher loop; release of prompts/hyperparameters/pseudocode aids reproducibility.
- Addresses real limitations of long-form scientific summarization (structure, coverage, factual fidelity) in a way that is backend-agnostic and transferable to other LLMs.

**Weaknesses:**

- Intro & positioning: Overly long and well-known historical context (extractive→abstractive→LLMs→RAG); also incorrectly describes BERT as generative. Should be tightened and corrected.
- Problem Definition (Section 3.1): Only a one-sentence restatement; consider folding into Section 3.2 or expanding with notation, desiderata (structure/fidelity/coverage), and evaluation targets.
- Faithfulness evaluation: No targeted factuality metrics (QA, entity-level consistency, or human judgment) despite claims of reduced hallucination and “human preference.” Needs structured factuality audits and error analysis.
- Structure evaluation mismatch: Claims of improved IMRaD-style coherence are not supported by overlap-based metrics. Consider section-aware ROUGE, discourse/section alignment metrics, or human annotation.
- Efficiency & cost: No reporting of inference time, token counts, or iteration depth—key given the repeated Teacher loop with k-NN retrieval and LLM passes. Need wall-clock per document vs. baselines, GPU hours, and scaling behavior.
- KG construction details: Missing specifics on extractors, supervision, accuracy, and graph statistics (nodes/edges per paper, error cases). Hard to assess the causal role of KG quality without this.
- Case study: The Attention Is All You Need example is illustrative but too limited to be representative.
- Chunking: Fixed chunking may break semantic boundaries (see "Semantic Self-Segmentation for Abstractive Summarization of Long Documents", AAAI 2022).

**Questions:**

1. KG pipeline: Which entity/relation extraction models are used, and is any supervision or fine-tuning involved?
2. Retrieval corpus & leakage: What corpus underlies the k-NN index? What is its domain coverage, size, and overlap with evaluation sets? Is there risk of pulling content too close to gold summaries?
3. Efficiency: What are the mean/median iteration counts, tokens generated/consumed, total latency, and cost relative to single-pass LLM prompting and supervised baselines?
4. Structure metrics: Can you provide section-aware or discourse-level metrics to substantiate structural-coherence claims?
5. Faithfulness audits: Will you add QA-based factuality metrics (entities, numbers, claims), contradiction rates, and a human study to support the fidelity/human-preference assertions?
6. Ablations: Can you include ablations on KG quality (noisy vs. gold graph), community detection variants (Leiden vs. alternatives), and retrieval-off settings to isolate the contribution of each component?
7. Baselines – training / fine-tuning: Are the encoder-decoder baselines (i.e., T5/LED/PEGASUS) evaluated in zero-shot mode, or are they fine-tuned on the same datasets? If fine-tuned, what are the training regimes and early-stopping criteria? If not fine-tuned, please justify comparability.

**Details Of Ethics Concerns:**

N/A.

---

### Official Review · Reviewer_eZ3M · 2025-10-31

**Soundness:** 1
**Presentation:** 3
**Contribution:** 2
**Rating:** 2
**Confidence:** 4

**Summary:**

This paper focuses on scientific paper summarization which is helpful to researchers by accelerating knowledge access. While arguing that the existing summarization approaches cannot capture the logical structure of scientific papers and may produce hallucination, This paper proposes a summarization framework inspired by the human summarization process, which is more structured and controlable than sequential processing. There are two modules: a student module generating the summary, and a teacher module assessing the summary. The student first writes a draft summary based on knowledge graphs and then get feedback from the teacher model based on reference-based reflection to improve the summary until the teacher module accepts it.

**Strengths:**

The idea of generating scientific paper summaries with knowledge graphs seems promising to capture more structural information to get better summaries.

**Weaknesses:**

I would be negative to this paper in an overall opinion. Here are my comments. I'd be very happy to discuss if the authors disagree.

1. My biggest concern is on evaluation for this paper. (1) Based on the problem definition in the first paragraph of the paper, scientific paper summaries should capture the IMRaD structure, condense complex content without omitting critical details, and ensure factual consistency. The framework is designed following this definition to generate the summary. However, abstracts are not always composed of IMRaD and there are usually high-level description while the evaluation is done on PubMed and arXiv using the abstracts as ground-truth summaries. The target summary in the paper is not aligned with the abstract provided in PubMed or arXiv. Therefore, evaluating generations based on those abstracts cannot well support the claims especially for logical structure. (2) There should be human evaluation for generated summaries on more recent scientific papers, as the metrics used in the paper are not good factuality and data contamination for DeepSeek and Qwen on PubMed and arXiv. There are other better factuality metrics, such as AlignScore.

2. Clams are not well supported by experiments. For example, there is not enough evidence to claim that the proposed approach outperforms other baselines in terms of human preference, without proper human evaluation experiments.

3. There have been a lot related summarization papers based on graphs, including pipeline and end-to-end approaches based on heterogeneous or homogeneous graphs, such as GraphSum and HGSum. The results in this paper should be compared with those papers based on other types of graphs at least in a qualitative way.

4. Implementation details of the framework is unclear. (1) how to get the nodes and edges for the graph is missing, and what level of information the nodes have is unclear (words, phrases, or description of Tasks, Methods, etc.). (2) how to generate the sub-summary for a sub-graph is missing, while this is an important part of the proposed framework. I assume that is is based on prompting, but I didn't find any prompt for this in the appendix. (3) what is the refernece corpus for the quality evaluation module?

5. There should be more baselines, including (1) simple iterative refinement based on LLMs and (2) simple LLM prompting. The experiments on T5, LED and PEGASUS are unnecessary because of their limited language capabilities.

Minor:

1. Please add more description about the two baseline approaches, SumCot and QA-prompting.

2. Add a reference to Section 3.4 in line 253 for the teacher feedback.

**Questions:**

Plese just give responses to the weaknesses.

**Details Of Ethics Concerns:**

None.

---

### Official Review · Reviewer_vgCj · 2025-10-31

**Soundness:** 2
**Presentation:** 3
**Contribution:** 2
**Rating:** 2
**Confidence:** 4

**Summary:**

The paper proposes ScholarSum, which is a pipeline for scientific paper summarization that tries to imitate how a student writes and a teacher critiques. The student side builds a knowledge graph from the paper, and the teacher side reads the draft. On ArXiv and PubMed the method beats T5, LED, PEGASUS and LLM prompting baselines on ROUGE, METEOR and BERTScore.

**Strengths:**

- The work grounds the summary in a knowledge graph so it can keep important facts like task, dataset, metric.

- The proposed method is benchmark on two scientific datasets and evaluated across multiple syntactical and semantic metrics.

**Weaknesses:**

- To run ScholarSum you must do entity and relation extraction, build a KG, run Leiden, generate sub summaries, run LLM for integration, then run a teacher that also does kNN retrieval. That is a lot of moving parts.

- The teacher compares the student output to abstracts from similar papers. That can bias the model toward generic biomedical or generic ML abstract style and away from the actual source paper, especially for niche or interdisciplinary papers.

- The baselines in the paper are limited. There are many strong models from 2024 and 2025 that could have been included for comparison. The authors should also test more recent and multiple LLMs to make the evaluation fair.

- The paper report ROUGE, METEOR, BERTScore. There is no human evaluation of factual errors or section alignment. For a paper that claims better factual fidelity that would be important.

- On ArXiv the differences over the better baselines are small. They are still wins, but not huge. So some of the benefit may be from better grounding on biomedical style text.

**Questions:**

- You set a max of 5 iterations. How often do you actually need more than 2? If you allow more iterations do you get better scores?

- What corpus is used for kNN retrieval in the teacher? what is "reference corpus" ?

- The thresholds 0.60 and 0.85 look task specific. How does the change affect the performance ? An ablation would have been better.

- Figure 2 seems to contain the OpenAI logo. If ChatGPT or another OpenAI model was used for prompting or generation, the paper should mention it clearly and also include direct comparisons with ChatGPT summaries.

---

### Official Review · Reviewer_j8iV · 2025-11-01

**Soundness:** 1
**Presentation:** 2
**Contribution:** 2
**Rating:** 2
**Confidence:** 4

**Summary:**

The paper proposes ScholarSum, a student-teacher framework designed for the abstractive summarization of complex scientific documents, especially for the article. The proposed framework is explicitly inspired by the human writing process, which involves drafting, reviewing, and revising. ScholarSum addresses challenges in scientific article summarization, namely capturing logical structure, avoiding information loss in long contexts, and ensuring factual fidelity. The system operates iteratively through two main modules: student module (knowledge graph + CoT reasoning) and the teacher module (KNN + revision). Experimental results on the ArXiv and PubMed datasets demonstrate that ScholarSum consistently outperforms strong baselines, producing summaries that are structurally coherent, factually comprehensive, and well aligned with human-written reference summaries.

**Strengths:**

- The proposed ScholarSum demonstrates robust quantitative superiority over baseline methods.
- The entire framework is explicitly inspired by the collaborative human writing process of drafting, reviewing, and revising.

**Weaknesses:**

- The framework integrates several established concepts, potentially diminishing its novelty.
- The proposed method benefits from strong pretrained generative models, iterative critique and revision techniques, and graph grounded retrieval, which is a combination of well-known methods in this field. While the specific application and structure are new, the core technological building blocks are combinations of widely adopted techniques.
- The proposed system involves a significant number of components, including KG construction, segmentation, two-stage generation, KNN algorithms, and revision methods. Although the paper presents systematic ablation studies confirming the importance of factual extractive grounding and the overall Teacher module, a deeper component-by-component analysis is necessary.
- The quantitative evaluation relies on traditional summarization models like T5 and PEGASUS and two specific LLM prompting methods (SumCot and QA-prompting). While ScholarSum outperforms these, I think that more sophisticated or carefully engineered prompting techniques for LLMs could yield methods competitive with ScholarSum.
- Usage of numerous components and iterative LLM calls suggests that the computational overhead may be substantial, particularly compared to single-pass summarization models. However, the paper lacks any analysis of the required computation cost, runtime, or efficiency required to achieve the reported results, leaving the practical deployability concerns unaddressed.

**Questions:**

- How's the computational cost for the proposed method?
- Could you provide ablation study for the proposed method?

---

### Note · Authors · 2026-01-05

I have read and agree with the venue's withdrawal policy on behalf of myself and my co-authors.